# Can Galectin-3 Be a Novel Biomarker in Chronic Lymphocytic Leukemia?

**DOI:** 10.3390/cells13010030

**Published:** 2023-12-22

**Authors:** Justyna Woś, Agata Szymańska, Natalia Lehman, Sylwia Chocholska, Michał Zarobkiewicz, Piotr Pożarowski, Agnieszka Bojarska-Junak

**Affiliations:** 1Department of Clinical Immunology, Medical University of Lublin, 20-093 Lublin, Poland; justyna.wos@umlub.pl (J.W.); agata.szymanska@umlub.pl (A.S.); 56730@student.umlub.pl (N.L.); michal.zarobkiewicz@umlub.pl (M.Z.); piotr.pozarowski@umlub.pl (P.P.); 2Department of Haematooncology and Bone Marrow Transplantation, Medical University of Lublin, 20-080 Lublin, Poland; sylwia.chocholska@umlub.pl

**Keywords:** CLL, Galectin-3, tumor microenvironment, prognostic markers

## Abstract

Galectin-3’s (Gal-3) effect on the pathogenesis of chronic lymphocytic leukemia (CLL) has not yet been extensively studied. The present study aims to analyze the potential role of Gal-3 as a prognostic biomarker in CLL patients. The Gal-3 expression was evaluated in CLL cells with RT-qPCR and flow cytometry. Due to the unclear clinical significance of soluble Gal-3 in CLL, our goal was also to assess the prognostic value of Gal-3 plasma level. Because cell survival is significantly affected by the interaction between Gal-3 and proteins such as Bcl-2, the results of Gal-3 expression analysis were also compared with the expression of Bcl-2. The results were analyzed for known prognostic factors, clinical data, and endpoints such as time to first treatment and overall survival time. Our research confirmed that Gal-3 is detected in and on CLL cells. However, using Gal-3 as a potential biomarker in CLL is challenging due to the significant heterogeneity in its expression in CLL cells. Moreover, our results revealed that Gal-3 mRNA expression in leukemic B cells is associated with the expression of proliferation markers (Ki-67 and PCNA) as well as anti-apoptotic protein Bcl-2 and can play an important role in supporting CLL cells.

## 1. Introduction

It is widely known that cancer cells and the surrounding tumor microenvironment (TME) have strong interactions and significant influence on each other. The milieu of chronic lymphocytic leukemia (CLL) is an essential element providing appropriate signals for survival, proliferation, and drug resistance [1]. Blood leukemic cells showing a high rate of spontaneous apoptosis after following ex vivo culture are an excellent example of the above. Numerous immune cells are present in the TME and play a significant role in supporting CLL cells [2]. Galactose-binding proteins known as galectins are key contributors in regulating tissue homeostasis in various cancers [3,4]. Consequently, galectins impact the progress of chronic inflammation in the tumor microenvironment [5]. They aid neoplastic cells in growing and surviving by regulating interactions with the extracellular environment and preventing anti-neoplastic immunity [3,5,6]. Galectins influence not only the transformation of cancer itself but also the formation of metastases, angiogenesis, and protection against apoptosis-inducing chemotherapy drugs [7,8]. Galectins play a variety of roles in different cancers, including hematological malignancies [9]. The tumor microenvironment’s critical component is Galectin 3 (Gal-3) [10]. It is expressed in several cell types and involved in physiological and pathological processes, such as cell adhesion and activation, chemoattraction, cell cycle, apoptosis, or cell growth and differentiation [11]. It should be emphasized that the function of Gal-3 depends on its subcellular location [12,13,14]. An intracellular Gal-3 acts as an anti-apoptotic factor, while an extracellular form acts mainly as a pro-apoptotic factor [13,14]. Moreover, there is a significant amino acid sequence similarity between Gal-3 and Bcl-2, a known blocker of apoptotic cell death [15,16]. The presence of Gal-3 in tumors and TME in various cancer types suggests that it contributes to immunosuppression and promotes tumor growth [11,17]. Gal-3 shows high expression in solid tumors, which is related to the stage of cancer advancement and indicates that Gal-3 plays an important role in the course of the disease [15]. High Gal-3 expression is also linked to poor prognosis in acute myeloid leukemia (AML) [9]. Among chronic B-cell lymphoproliferative diseases (B-CLPD), high levels of Gal-3 are observed in patients with diffuse large B-cell lymphoma (DLBCL) and multiple myeloma (MM) [18]. Intracellular and extracellular mechanisms of Gal-3 supporting the survival and proliferation of CLL cells indicate that this protein is also an important component of the TME and may be a possible target for therapy [10].

The most clinically significant markers of poor CLL outcome are the lack of mutations in the immunoglobulin heavy chain variable region (IGHV), chromosomal abnormalities, as well as high expression of ZAP-70 (zeta-associated protein 70) and CD38 [19,20,21]. Chromosomal alterations in CLL are detected in up to 80% of patients. Among them, deletions of 11q (found in 5–20% of cases), 13q (found in more than 50% of CLL patients), 17p (3–8%), and trisomy 12 (10–20%) have a known prognostic value and play an important role in CLL pathogenesis and evolution, determining patients outcome and therapeutic strategies [22].

The present study aims to analyze the potential role of Gal-3 as a prognostic biomarker in CLL patients. We evaluated the expression of Gal-3 in CLL cells using RT-qPCR and flow cytometry. As the clinical significance of the soluble form of Gal-3 (sGal-3) in CLL patients is still unclear, our goal was also to assess sGal-3 prognostic value. Because cell survival is significantly affected by the interaction between Gal-3 and proteins such as Bcl-2 [23], the results of Gal-3 expression analysis were also compared with the expression of Bcl-2. We are the first team to correlate Gal-3 mRNA expression levels with the proliferation markers (i.e., Ki-67 and PCNA) of CLL cells. The results obtained were analyzed for known prognostic factors, clinical data, and endpoints such as time to first treatment (TTFT) and overall survival time (OS).

## 2. Materials and Methods

### 2.1. Study Material and Patients’ Characteristics

The peripheral blood from 146 CLL patients and 26 healthy volunteers (HV) was collected after signing an informed written consent to participate in the research.

Peripheral blood (PB) samples were obtained from patients who had not received prior therapy for CLL. Eight CLL patients requiring treatment were studied at two time points: before the start of the treatment and 6 or 12 months after therapy. The study and control group recruitment took place in the Department of Hematooncology and Bone Marrow Transplantation of the Medical University of Lublin, Poland. The control group was selected based on age and sex, complementary to the patients. The characteristics of cohorts are presented in Table 1. For CLL patients, the following criteria had to be met: the diagnosis based on standards from the International Workshop on Chronic Lymphocytic Leukemia [21], lack of any other cancer or bone marrow disease, no autoimmune disease, and no circulating and/or pulmonary system failure. The clinical CLL stage was determined according to the Rai classification system [24]. Cytogenetic abnormalities such as trisomy 12 and deletions of 13q, 17p, and 11q were assessed. Cytogenetic studies were conducted either during diagnosis or before treatment. Detailed information is provided in Table 1. Additionally, we summarized the morphology and basic immune profiling data (Table 1). These include prognostic factors used worldwide, like ZAP-70 and CD38 expression on leukemic cells. The study protocol was conducted according to the Declaration of Helsinki and approved by the Local Bioethics Committee.

### 2.2. Flow Cytometry Assessment

#### 2.2.1. PBMCs Isolation

Blood collected from tubes with EDTA was used for peripheral blood mononuclear cells (PBMCs) isolation in a density gradient. For this purpose, the Gradisol L (Cat No.: 9003.1, Aqua-Med, Łódź, Poland) was utilized. PBMCs were used in Section 2.2.2, Section 2.3, Section 2.4.1 and Section 2.6.

#### 2.2.2. Cell Surface and Intracellular Stainings

PBMCs were stained with antibodies anti-CD19 FITC (Clone HIB19, Cat No.: 302206, BioLegend, SanDiego, CA, USA) and anti-Galectin-3 PE (Clone B2C10, Cat No.: 565676, BD Biosciences, Franklin Lakes, NJ, USA). B lymphocytes with Gal-3 membrane expression were analyzed within CD19+ cells (Figure 1A). For intracellular assessment of Gal-3 expression, the Fc receptor was blocked (FcR Blocking Reagent, Cat No. 130-059-901, Miltenyi Biotec, Bergisch Gladbach, Germany) according to manufacturer instructions. Then, lymphocytes were fixed and permeabilized with Cytofix/Cytoperm Fixation/Permeabilization Kit (Cat No. 554714, BD Biosciences, Franklin Lakes, NJ, USA). Next, cells were stained with monoclonal antibodies against Gal-3 and incubated for one hour in darkness. The acquisition of all probes was performed on CytoFlex LX (Beckman Coulter, Brea, CA, USA), and then cytometric data were analyzed with Kaluza Analysis (Beckman Coulter, CA, USA).

### 2.3. Confocal Microscopy

Multilabel confocal microscopy was performed on peripheral blood mononuclear cells using antibodies to CD19 and Gal-3 to visualize the intracellular localization of Gal-3 in CD19+ cells. It came to our attention that many antibodies used in flow cytometry can also be applied in fluorescence microscopy. In some experiments, we took a portion of intracellular-stained cells for flow cytometry analysis (as described above) and immobilized them onto microscope slides using cytocentrifugation. Cells mounted to microscope slides were then stained with DAPI (1 µg/mL, ThermoFisher Scientific, Waltham, MA, USA; Cat No. D1306) for 15 min at room temperature to show their nuclei. Confocal imaging was performed using a Nikon A1R confocal microscope (Nikon, Tokyo, Japan). FITC fluorescence was excited by a 488 nm argon laser and measured using a 525 nm spectral filter. 561 and 405 nm lasers and 595 and 450 nm filters were applied for PE and DAPI, respectively.

### 2.4. Quantitative Measurement of Gal-3 mRNA Expression

#### 2.4.1. Magnetic Isolation of CD19+ Cells

Leukemic cells were separated from PBMCs with the MACS system (Miltenyi Biotec, Bergisch Gladbach, Germany). The positive isolation was performed. First, cells were washed in MACS buffer and then incubated with FcR blocking reagent (Cat No. 130-059-901, Miltenyi Biotec, Bergisch Gladbach, Germany). Next, following the incubation with CD19 MicroBeads (Cat No. 130-050-301, Miltenyi Biotec, Bergisch Gladbach, Germany), cells were resuspended in MACS buffer. Then, they were placed in the column and washed three times. Finally, the column was taken from the magnetic field, and B lymphocytes were pushed out. The purity of achieved cells was greater than 95%, as assessed by flow cytometry. Isolated B lymphocytes were centrifuged at 300× *g* (10 min), and the pellet was collected and cryopreserved at −80 °C until processing for RT-qPCR.

#### 2.4.2. RT-qPCR

In the beginning, total RNA from purified B lymphocytes was isolated using the QIAamp RNA Blood Mini Kit (Cat No. 52304, Qiagen, Inc., Valencia, CA, USA). Next, the reverse transcription was performed with the QuantiTect Reverse Transcription kit (Cat No. 205311, Qiagen, Inc., Valencia, CA, USA). The TaqMan Gene Expression Master Mix (Thermo Fisher Scientific, Applied Biosystems, Inc., Waltham, MA, USA) was used, and qPCR reactions were run on the 7300 Real-Time PCR System (Applied Biosystems). The probes for *LGALS3* (Hs03680062_m1), *BCL2* (Hs04986394_s1), *Ki-67* [Hs01032437_m1], and *PCNA* [Hs00427214_g1] genes expression analysis were used with *GAPDH* (Cat No. 4310884E, Thermo Fisher Scientific, Waltham, MA, USA) as a housekeeping gene. Gene expression levels were determined using the following formula: 2^−ΔCq^ when the ΔCq was Cqt (target gene)—Cqr (housekeeping gene) [25]. The qPCR was carried out in duplicates.

### 2.5. Gal-3 Detection in Plasma

Plasma (EDTA) was used in the assay. Gal-3 concentration was determined using enzyme-linked immunoassay (ELISA) (Galectin 3 Human ELISA Kit, Cat No. BMS279-4, ThermoFisher Scientific, Waltham, MA, USA) according to the manufacturer’s instructions. The experiment was performed in duplicates. The optical density was measured at 450 nm wavelength with the ELISA Reader Viktor3 (Perkin Elmer, Waltham, MA, USA). The detection limit for Gal-3 was 0.29 ng/mL.

### 2.6. Fluorescent In Situ Hybridization (FISH)

The Vysis CLL FISH Probe Kit (Abbott GmbH, Wiesbaden, Germany) was used to detect common genomic aberrations: trisomy 12 and deletions of 11q22.3 (*ATM*), 17p13.1 (*TP53*), and 13q14.3. The FISH protocol was described previously [26].

### 2.7. Statistical Analysis

Statistica 13.1 software (StatSoft, Cracow, Poland) and GraphPad Prism 9 (GraphPad Software, San Diego, CA, USA) were used for statistical analysis. The D’Agostino-Pearson test was used to determine the data distribution. For the comparison between groups, the U Mann–Whitney, Kruskal–Wallis, or Wilcoxon’s matched-pairs rank tests were conducted. The results are presented as median with interquartile range (IQR). The Spearman rank correlation coefficient was used for correlation testing. The overall survival (OS) and time to first treatment (TTFT) were presented as Kaplan–Meier curves. Patients were divided into two groups based on Gal-3 concentration or expression. The cut-off points were selected based on the receiver operating characteristics (ROC) analysis. The Youden index and AUC (area under the ROC curve) were also estimated. The hazard ratios (HRs) were calculated using Cox proportional hazard models, both with and without the use of adjustments for multiple variables. The *p*-value smaller than 0.05 was considered statistically significant.

## 3. Results

### 3.1. Heterogenous Gal-3 Expression in CLL Patients

Gal-3 expression can be found in the nucleus, cytoplasm, and cell surface. It can also be secreted into the extracellular matrix and then circulation [11]. Based on that, we evaluated the expression of Gal-3 on the CD19+ cells surface, the expression of the *LGALS3BP* gene at the mRNA level, and the Gal-3 plasma concentration. In addition, in 24 CLL patients, the expression of Gal-3 was analyzed both on the CD19+ cell surface and intracellularly. B lymphocytes with Gal-3 membrane expression were analyzed within CD19+ cells (Figure 1A). Membrane Gal-3 expression did not differ significantly from its intracellular expression (median [IQR], 2.890 [1.270–6.050]% vs. 4.670 [1.440–9.260]%, *p* > 0.05, Figure 1B). Representative dot plots of three CLL patients show heterogeneous Gal-3 expression on CD19+ cells (Figure 1C). Flow cytometry analysis showed significant surface and intracellular Gal-3 expression within B CD19+ cells. Confocal microscopy confirmed that Gal-3 was expressed both in the cytoplasm and nucleus of CD19+ cells (Figure 2).

We did not demonstrate any significant differences in the percentage of Gal-3-positive B cells in CLL patients compared to the control group (Figure 3A). We also analyzed purified CD19+ cells for Gal-3 mRNA expression by RT-qPCR. We observed significantly higher (*p* < 0.05) expression of Gal-3 mRNA in CLL patients than in the healthy volunteers (Figure 3B). Plasma levels of Gal-3 were also higher in CLL patients than in the control group. However, the difference was not statistically significant (Figure 3C).

Plasma levels of Gal-3 were not in correlation with leukocytosis and lymphocytosis. Likewise, membrane expression of Gal-3 and at the mRNA level did not correlate with these parameters. The concentration of Gal-3 was in a weak positive correlation with the serum level of β2-microglobulin (r = 0.18; *p* < 0.05). We did not detect any correlation between a form of Gal-3 expression and the age and sex of the studied patients. We have shown that a higher level of Gal-3 correlated with the clinical stage of the disease according to the Rai classification. We grouped 146 patients according to the Rai stage (ranging from 0 [low risk] through I or II [intermediate risk] to III or IV [high risk]). A significantly higher plasma level of Gal-3 was found at stage III/IV than at stage 0. Importantly, a significantly higher Gal-3 level was found in patients at stage III/IV compared to the control group (Figure 4A). Moreover, a significant increase has been detected in the percentage of Gal-3-positive B cells in patients with III/IV stage disease as compared to stage 0 and stage I/II and healthy volunteers (Figure 4B). The analysis at the mRNA level confirmed these results (Figure 4C).

Taking into account adverse prognostic factors, i.e., ZAP-70 and CD38, we observed no statistically significant (*p* > 0.05) differences in Gal-3 concentration (Figure 5A,B). On the contrary, the ZAP-70-positive group of CLL patients had a significantly (*p* < 0.001) higher percentage of B lymphocytes expressing Gal-3 compared to the ZAP-70-negative group (Figure 5C). However, in the case of CD38, the observed difference was not statistically significant (Figure 5D). For ZAP-70, the study of the Gal-3 mRNA expression levels confirmed the results of cytometric analysis (Figure 5E). However, it is worth noting that Gal-3 mRNA expression in the CD38+ group is higher than in the HV group (Figure 5F). Increased CD38 expression in CLL cells is associated with aggressive disease [19,21]. It is suggested that in CLL patients, the CD38+ group is characterized by generally observed high Gal-3 expression.

All three methods of Gal-3 expression measurement presented a similar significant tendency for patients with different statuses of IGHV mutation, which corresponded with an increased expression level of the Gal-3 in patients with unmutated IGHV genes when compared to HV (Figure 6A–C). In addition, CLL patients with unmutated IGHV genes display a higher percentage of Gal-3 positive B cells (Figure 6B) and Gal-3 mRNA expression in B cells (Figure 6C) than those with mutated IGHV genes (*p* < 0.05).

### 3.2. The Relationship between the Molecular Profile of Patients and Gal-3 Levels

All three methods of Gal-3 expression measurement presented a similar significant tendency for sole 13q deletion, which corresponded with a reduced expression level of the Gal-3 when compared to patients carrying del(17p), del(11q), and/or 12 trisomy (Figure 7). These changes were especially visible for Gal-3 expression on leukemic B cells measured using the flow cytometry. It is worth noting that the cohort with del(17p) along with del(11q) and/or trisomy 12 had statistically the highest levels of Gal-3 (Figure 7).

### 3.3. Assessment of the Influence of Gal-3 Expression on the Clinical Outcomes

During the follow-up period, 58 patients required treatment. Details of the treatment are provided in Table 1. The criteria of indications for treatment and response assessment proposed by the International Workshop on Chronic Lymphocytic Leukemia (IWCLL) were used [21]. The sGal-3 concentration (Figure 8A), percentage of B cells with surface expression of Gal-3 (Figure 8B), and Gal-3 mRNA expression (Figure 8C) measured at the time of diagnosis were higher in patients requiring therapy during the observation period when compared to other patients or healthy volunteers. A statistically significant difference between patients requiring treatment and those who did not need therapy was revealed only in the case of Gal-3 mRNA expression (*p* < 0.05; Figure 5C).

The group of treated patients was assessed with regard to therapy outcomes (Figure 9). The patients were classified as responders (those who obtain complete or partial remission (CR, PR)) and non-responders (those who have stable disease (SD) or progression (PD)). Statistical analysis indicated that CLL cells with surface Gal-3 expression were not a predictor of therapy outcome (Figure 9B). However, variability in treatment regimens used could influence the results. On the contrary, the non-responding group had significantly higher Gal-3 plasma levels (Figure 9A) and Gal-3 mRNA expression (Figure 9C) in CLL cells compared to the patients with CR/PR.

Based on the ROC analysis (Figure 10A–C), we determined the optimal thresholds for three forms of Gal-3 that could discriminate CLL cases with del(17p) and/or del(11q) from CLL patients without unfavorable aberrations. The cut-off value for sGal-3 concentration was >12.92 ng/mL (Figure 10A), the percentage of Gal-3-positive B cells was >6.0% (Figure 10B), and for Gal-3 mRNA expression in B cells was >3.392 (Figure 10C). Subsequently, the group of CLL patients was divided into low and high groups based on the established cut-off values. The Kaplan–Meier test showed that there was no statistically significant difference in the prediction of TTFT between groups with high and low concentrations of Gal-3 (Figure 10D). On the other hand, we observed that a high percentage of Gal-3+/CD19+ cells (>6.0%) was associated with shorter TTFT (Figure 10E). Similarly, Kaplan–Meier analysis showed a statistically significant difference between a group of patients with high versus low expression of Gal-3 mRNA (Figure 10F). In univariate analysis, predictors of TTFT included the presence of del(17p)/del(11q), a high β2M level, high ZAP-70 expression, high Gal-3 mRNA expression, and a high percentage of Gal-3 positive B cells (Table 2). However, in multivariate analysis, only high β2M level and the presence of del(17p)/del(11q) were found to be significant predictors of TTFT (Table 2).

Nevertheless, in univariate and multivariate analysis, OS (overall survival) seemed not to be influenced by the concentration of Gal-3, percentage of Gal-3+CD19+ cells, and Gal-3 mRNA expression in leukemic B cells (*p* > 0.05).

Eight patients requiring treatment were examined at two time points: at the time of diagnosis and 6 or 12 months after therapy. Gal-3 expression assessed over time was presented in Table 3. We noted that the expression of Gal-3 in individual patients changed over time and was significantly higher at the time of diagnosis than after therapy (*p* < 0.01; Table 3).

### 3.4. The Relationship between the Level of mRNA Expression of Gal-3 and Bcl-2 and Proliferation Markers (Ki-67 and PCNA)

Current evidence suggests that GAL-3 is a molecule with strong anti-apoptotic activity in many types of cancer cells and plays an important role in promoting cancer proliferation [27]. That is why we decided to assess whether there is a relationship between the level of Gal-3 expression in leukemic cells and the expression of Bcl-2 and proliferation markers (Ki-67 and PCNA).

In the group of CLL patients, there was a weak positive correlation between the Gal-3 mRNA and Ki-67 (r = 0.297; *p* < 0.01) and a moderate correlation with PCNA mRNA expression (r = 0.514; *p* < 0.01). Moreover, a moderate positive (r = 0.481; *p* < 0.01) correlation with Bcl-2 mRNA level in purified CD19+ cells (r = 0.481; *p* < 0.01) was found. Then, the group of CLL patients was divided into Gal-3^low^ (<3.392) and Gal-3^high^ (>3.392) groups. The group with Gal-3 mRNA level > 3.392 in B cells showed significantly higher mRNA expression levels of Bcl-2 (Figure 11A), Ki-67 (Figure 11B), and PCNA (Figure 11C).

## 4. Discussion

In this study, we examined the expression of Gal-3 in B cells. Using flow cytometry and RT-qPCR, we demonstrated higher expression of Gal-3 mRNA in CLL patients than in the healthy volunteers. Different results were obtained by Asgarian-Omran et al. [28], who showed reduced Gal-3 expression at the mRNA level in CLL patients. They studied Gal-3 expression in total PBMCs [28], while we examined Gal-3 mRNA expression in purified CD19+ cells. It should be noted that Gal-3 is found in many cells, including monocytes, macrophages, dendritic, B and T cells [29]. Furthermore, in our study, quantitative RT-PCR (RT-qPCR) was used; in turn, Asgarian-Omran et al. [28] performed RT-PCR and classical gel electrophoresis, and they calculated the ratios of the galectin band to that of β-actin.

Gal-3 can be found in the nucleus and cytoplasm as well as on the cell surface. It can also be secreted into the extracellular matrix and then enter circulation [11]. Based on that, we also evaluated intracellular and membrane expression of Gal-3 on the CD19+. Surface and intracellular Gal-9 expression was heterogeneous in CLL patients. Nevertheless, we have not observed significant differences between these forms of Gal-3 expression. Further analysis showed no significant differences in the percentage of Gal-3 positive B cells in CLL patients compared to controls. Our results are consistent with previous reports of Michalová et al. [30], who also analyzed Gal-3 expression in CLL patients using flow cytometry. They observed no differences in Gal-3 surface expression between the CLL group and normal B cells. On the other hand, Michalová et al. [30] showed overexpression of intracellular Gal-3 in CLL patients compared to normal B cells. It should be noted that in our research, we used purified CD19+ B cells from healthy donors as a control. On the contrary, Michalová et al. [30] applied normal mature B-cells identified as polyclonal CD5^neg^/CD20^high^CD19^+^ cells from CLL patients. It is worth noting that, as mentioned above, Gal-3 expression is found in the nucleus, cytoplasm, and cell surface [10,11]. However, its location is strongly related to various factors, such as the type of a cell and the state of proliferation, transformation, or tumor progression, and the localization of Gal-3 in the cell itself determines its functions [16]. This can explain the difficulty in determining the cellular origin of the galectin protein in and on leukemic cells. In their research, Fei et al. [31] indicated that Gal-3, which was detected in and on acute lymphoblastic leukemia (ALL) cells, derived from stromal cells, which expressed the protein on their surface and secreted it in a soluble form [31]. ALL cells internalize soluble and stromal-bound Gal-3, which is then transported to the nucleus and stimulates the transcription of endogenous *LGALS3* mRNA [31]. However, to our knowledge, no analogous studies have been performed in CLL. Nevertheless, our analysis may suggest that Gal-3 assessed on B cells mainly derives from endogenous synthesis. It is worth noting that in all CD19+ cells, we found mRNA expression for Gal-3.

In the current study, we did not demonstrate any significant differences in plasma concentration of Gal-3 in CLL patients in comparison with healthy controls. On the contrary, Wdowiak et al. [32] observed a reduction in Gal-3 serum levels in CLL patients compared with healthy controls. In our study, higher plasma levels of Gal-3 correlated with the clinical stage of the disease according to the Rai classification. Importantly, a significantly higher Gal-3 level was found in patients at stage III/IV compared to the control group. Likewise, the percentage of Gal-3-positive B cells correlated with the clinical stage of the disease. The analysis at the mRNA level confirmed these results. High Gal-3 expression correlated positively with poor prognosis in CLL patients, which suggests that this molecule might be a prognostic biomarker of CLL. High expression of ZAP-70 and CD38 are well-known adverse prognostic factors [33,34]. We demonstrated significantly higher Gal-3 expression in ZAP-70+ and CD38+ patients compared to the control group. It is worth noting that these molecules are not the only prognostic factors characteristic of CLL [34,35]. Further attention is paid to various markers, such as the presence of mutations in the immunoglobulin variable gene (IGHV) and chromosomal aberrations (such as deletion of 17p, 11q, and 13q), which are used to assess the prognosis and adjust the treatment. In this study, we observed reduced Gal-3 levels in patients with 13q deletion, which is associated with a good prognosis compared to patients with del(17p), del(11q), and/or 12 trisomy. These changes were particularly visible in the case of Gal-3 expression on leukemic B cells as measured using flow cytometry. Michalová et al. [30] achieved comparable results. They found that cytoplasmic Gal-3 was predominantly overexpressed in patients with 17p deletion compared to those without 17p−. They did not find any connections between Gal-3 expression and the presence of 11q−, 13q−, and 12+ [30]. It should be noted that Stiasny et al. [36] showed that Gal-3 expression is largely suppressed in cells with TP53 mutations, which may explain increased levels of Gal-3 expression in TP53-mutated tumors. CLL patients with unfavorable chromosomal aberrations, such as del(17p) or del(11q), are usually more likely to have more rapid disease progression and poorer response to treatment [37]. We observed that a high percentage of Gal-3+/CD19+ cells and high mRNA Gal-3 expression were associated with a shorter time to first treatment. It is worth mentioning that cytoplasmic Gal-3 expression appears to be strongly linked to an anti-apoptotic function and drug resistance in cancer cells [38]. Wdowiak et al. [32] observed significantly higher Gal-3 levels in patients who succumbed to the disease compared to patients with CR, PR, SD and PD, with the PD patients showing the lowest Gal-3 concentration. On the contrary, in our study, the non-responding group had significantly higher Gal-3 plasma level and Gal-3 mRNA expression in B cells compared to the patients responding to treatment (complete or partial remission). However, the percentage of CLL cells with surface Gal-3 expression was not a predictor of therapy outcome. Variability in treatment regimens used could influence the results. We observed that Gal-3 expression changed over time and was significantly higher before the initiation of therapy when compared to the values after therapy. There was a decline in Gal-3 expression after treatment. However, a small number (n = 8) of patients is what limits our study. The results of CLL patients are promising, but larger trials with longer follow-up are necessary. Nonetheless, it appears that the expression of Gal-3 increases with the disease progression and decreases after therapy.

Due to the significant heterogeneity in its expression in CLL cells, using Gal-3 as a potential biomarker in CLL can be challenging. Thijssen et al.’s [39] review presented data on Galectin-3 and other members of the Galectin family as prognostic factors for various malignancies. It is crucial to note that changes in galectin expression are often linked to classic prognostic markers such as tumor grade or stage [39]. This was also demonstrated in our study of CLL patients.

Because cell survival is significantly affected by the interaction between Gal-3 and proteins such as Bcl-2 [23], the results of Gal-3 mRNA expression analysis were also compared with the mRNA expression of Bcl-2. It is common knowledge that the anti-apoptotic Bcl-2 protein is overexpressed in leukemic cells in all CLL patients [40,41,42]. In our study, we demonstrated a positive correlation between Gal-3 and Bcl-2 mRNA expressions. It is worth recalling that, in the past, it was thought that the slow accumulation of CLL cells was caused by defective apoptosis rather than proliferation [43,44]. However, that perspective has changed. In vivo, CLL cells displayed significant continuous proliferation [44]. It is important to note that malignant B cells are dependent on survival signals from non-neoplastic cells nearby [45]. Current evidence also suggests that Gal-3 is not only a molecule with strong anti-apoptotic activity, but it also plays an important role in promoting the proliferation of many types of cancer cells [27]. That is why we decided to see whether there is a relationship between the level of Gal-3 expression in leukemic cells and the expression of proliferation markers (Ki-67 and PCNA). The group of patients with a high Gal-3 mRNA expression in CLL cells showed significantly higher mRNA expression levels of Ki-67 and PCNA. It has to be remembered that Gal-3 expression is dependent on the cell cycle. On the other hand, it can also impact the advancement of the cell cycle [16]. In the study by Giglio et al. [46], the expression of PCNA in CLL samples was higher than that in normal PBMCs. Moreover, high levels of PCNA expression by unstimulated CLL cells at the time of diagnosis may identify patients who have poorer prognosis due to higher proliferative activity [47]. Although a direct interaction between Gal-3 and PCNA or Ki-67 has not been reported in the literature, their roles in promoting cell proliferation and cancer progression suggest potentially interconnected pathways in cancer biology. Of note, we are the first team to correlate Gal-3 mRNA expression levels with the proliferation markers (i.e., Ki-67 and PCNA) of CLL cells. We have taken into account these molecules also because it has been shown that the upregulation of Gal-3 and PCNA reduces tumor cell lysis by NK cells [48,49].

## 5. Conclusions

We acknowledge that our observations cannot fully define Gal-3’s biological role in CLL. Our research confirmed that Gal-3 is detected in and on CLL cells. Moreover, high mRNA levels suggest that Gal-3 is directly expressed by CLL B cells. However, using Galectin-3 as a potential biomarker in CLL is challenging due to the significant heterogeneity in its expression in CLL cells. Overall survival or time to first treatment in CLL patients cannot be predicted using Gal-3 expression. Our results revealed that Gal-3 mRNA expression in leukemic B cells is associated with the expression of proliferation markers (Ki-67 and PCNA) as well as anti-apoptotic protein Bcl-2 and can play an important role in supporting CLL cells.

## Figures and Tables

**Figure 1 cells-13-00030-f001:**
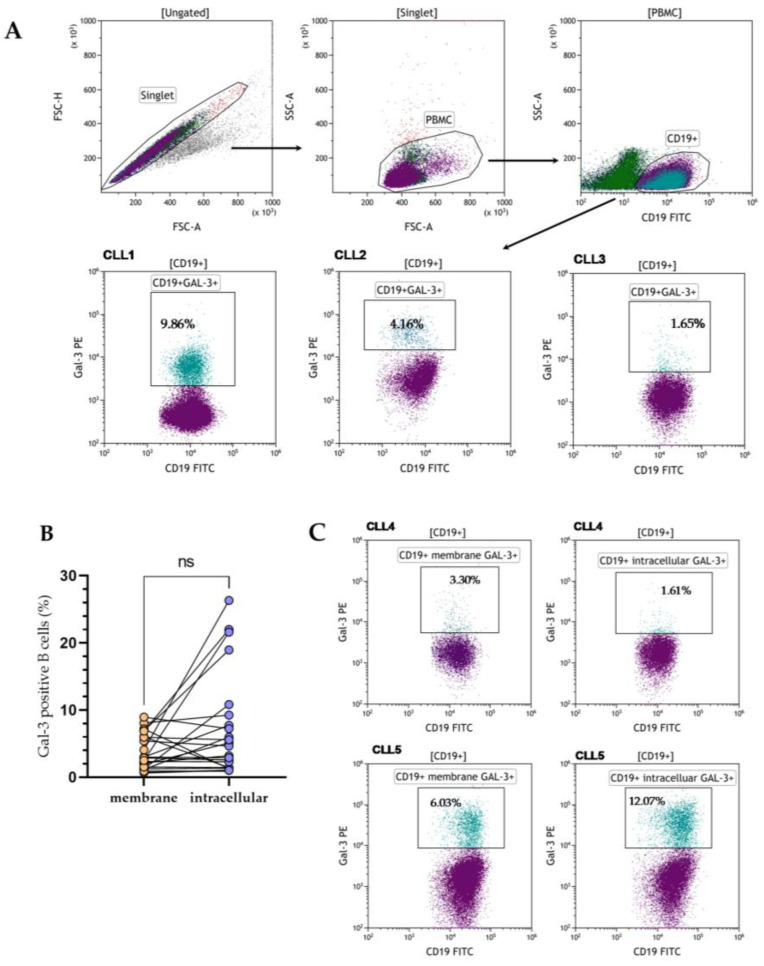
The flow cytometry data (dot plots) showed the gating strategy used to identify Gal-3 positive CD19+ cells. The input gate is displayed in the title of each dot plot (**A**). Representative dot plots of three CLL patients (CLL1–CLL3) with different levels of membrane Gal-3 expression. Data were analyzed using Kaluza 2.1.1 software (Beckman Coulter, Brea, CA, USA) (**A**). Percentage of CD19+ cells with surface and intracellular Gal-3 expression in 24 CLL patients (**B**). Representative dot plots of two CLL patients (CLL4–CLL5) with identification of CD19+ cells and with membrane and intracellular Gal-3 expression (**C**). FSC, forward scatter; SSC side scatter; PBMC, peripheral blood mononuclear cells; ns, not significant.

**Figure 2 cells-13-00030-f002:**
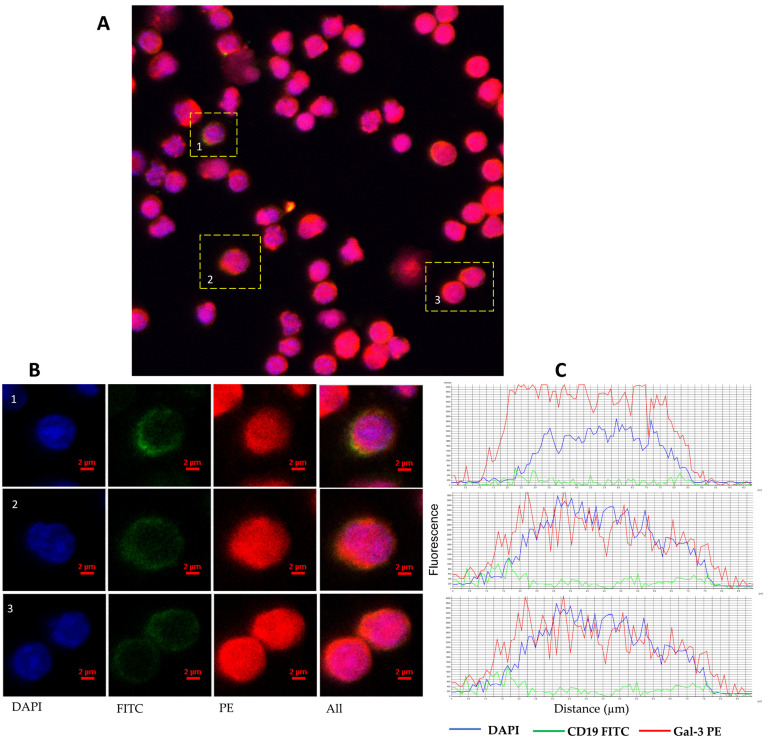
Representative confocal microscopy images of Gal-3 of CLL patient. Confocal microscopy confirmed intracellular expression of Gal-3 in CD19+ B cells. (**A**) Picture shows a confocal microscopy scanning area with overlaying DAPI, FITC, and PE fluorescence. (**B**) Images of representative cells were shown as individual fluorescent channels and merged ones. Blue represents the nuclear staining of DNA (DAPI), green is the superficial staining of CD19 (green), and red is the cytoplasmic staining of Gal-3 (PE). (**C**) Pictures show the intensity of cellular profiles of the fluorescence of DNA, CD19, and Gal-3. Confocal imaging was performed using a Nikon A1R confocal microscope (Tokyo, Japan).

**Figure 3 cells-13-00030-f003:**
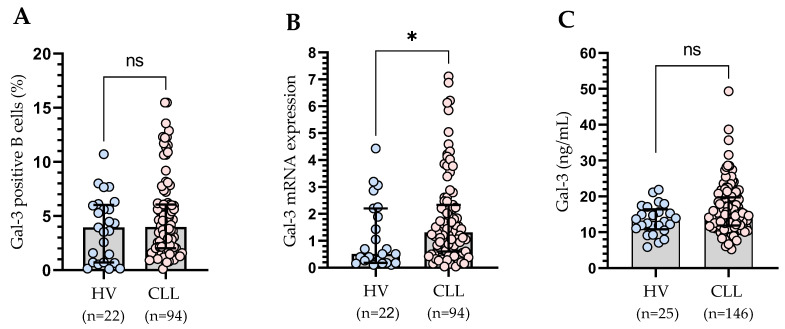
Percentage of Gal-3-positive CD19+ cells in CLL patients and healthy controls (**A**). Gal-3 mRNA expression in B cells of CLL patients and control groups (**B**). Serum Gal-3 concentration of CLL patients and control groups (**C**). The bar plot is used to show the median. “Whiskers” represent the lower (Q1) and upper (Q3) quartiles. The Mann–Whitney U test was used for the comparative analysis of variables. HV, healthy volunteers; ns, not significant; * *p* < 0.05.

**Figure 4 cells-13-00030-f004:**
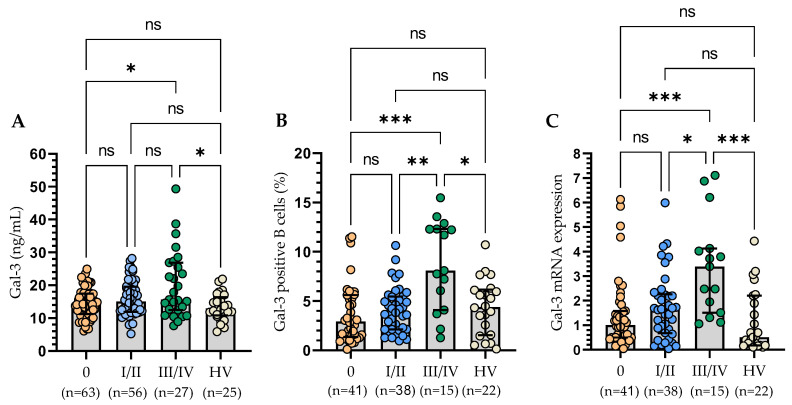
Serum Gal-3 concentration of CLL patients according to the Rai stages (**A**). Gal-3 expression on CD19+ cells in CLL patients in different disease stages (**B**). Gal-3 mRNA expression in B cells of CLL patients in different disease stages (**C**). The bar plot was used to show the median. “Whiskers” represent the lower (Q1) and upper (Q3) quartiles. The Kruskal–Wallis test with Dunn correction was used for comparative analysis of the variables. ns, not significant; * *p* < 0.05, ** *p* < 0.01, *** *p* < 0.001.

**Figure 5 cells-13-00030-f005:**
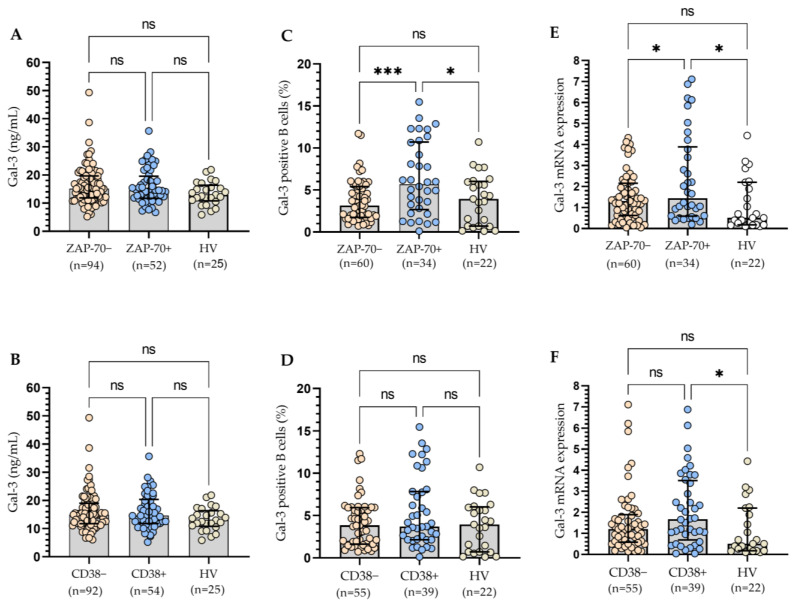
Plasma Gal-3 concentration in ZAP-70+/ZAP-70− (**A**) and CD38+/CD38− CLL patients (**B**). Gal-3 expression on CD19+ cells in ZAP-70+/ZAP-70− (**C**) and CD38+/CD38− CLL patients (**D**). Gal-3 mRNA expression in B cells in ZAP-70+/ZAP-70− (**E**) and CD38+/CD38− CLL patients (**F**). The bar plot was used to show the median. “Whiskers” represent the lower (Q1) and upper (Q3) quartiles. The Kruskal–Wallis test with Dunn correction was used for comparative analysis of the variables. ns, not significant; * *p* < 0.05, *** *p* < 0.001.

**Figure 6 cells-13-00030-f006:**
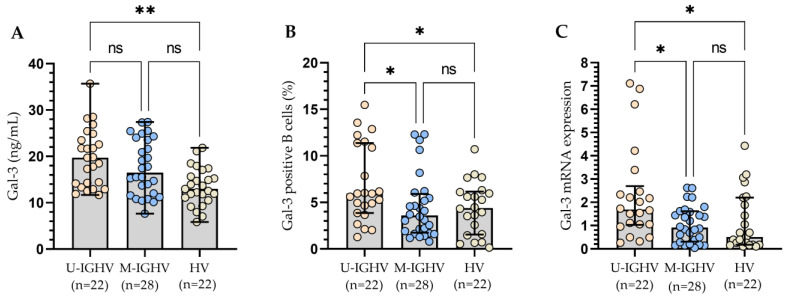
Plasma Gal-3 concentration (**A**), Percentage of Gal-3 positive B cells (**B**), and Gal-3 mRNA expression in B cells (**C**) from CLL patients with different status of IGHV mutation. The bar plot was used to show the median. “Whiskers” represent the lower (Q1) and upper (Q3) quartiles. The Kruskal–Wallis test with Dunn correction was used for comparative analysis of the variables. * *p* < 0.05, ** *p* < 0.01, ns, not significant; IGHV, immunoglobulin heavy chain variable gene; U-IGHV, unmutated IGHV; M-IGHV, mutated IGHV.

**Figure 7 cells-13-00030-f007:**
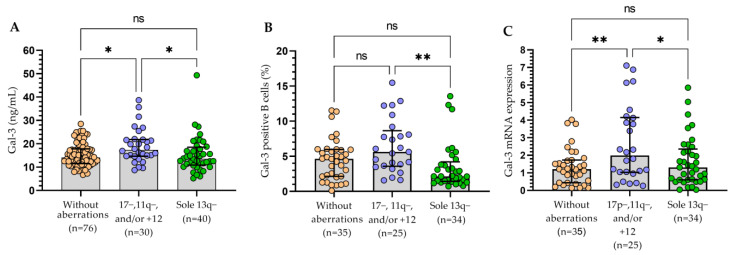
Gal-3 in CLL patients with chromosomal abnormalities (17p−, 11q−, +12, and 13q−). Plasma level of Gal-3 (**A**). Percentage of Gal-3 positive B cells (**B**). Gal-3 mRNA expression (**C**). The bar plot is used to show the median. “Whiskers” represent the lower (Q1) and upper (Q3) quartiles. The Kruskal–Wallis test with Dunn correction was used for comparative analysis of the variables. ns, not significant; * *p* < 0.05, ** *p* < 0.01.

**Figure 8 cells-13-00030-f008:**
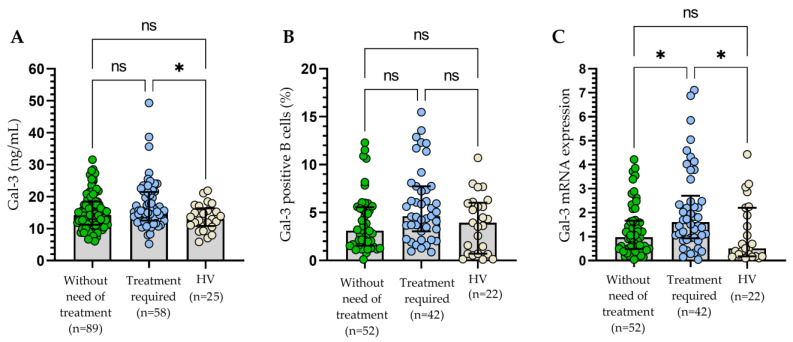
Concentration of Gal-3 (**A**), percentage of Gal-3+ B cells (**B**), and Gal-3 mRNA expression (**C**) in patients requiring therapy and those who did not need CLL therapy during the observation period. The bar plot is used to show the median. “Whiskers” represent the lower (Q1) and upper (Q3) quartiles. The Kruskal–Wallis test with Dunn correction was used for comparative analysis of the variables. ns, not significant; * *p* < 0.05.

**Figure 9 cells-13-00030-f009:**
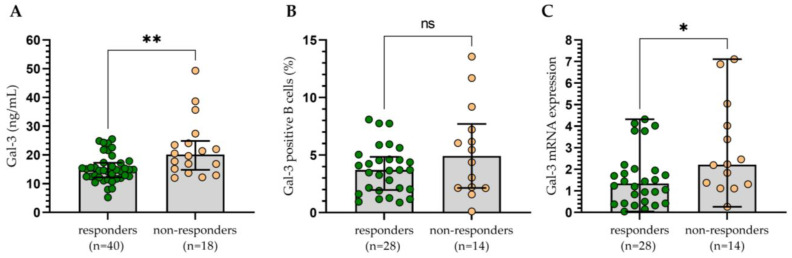
The concentration of Gal-3 (**A**), percentage of Gal-3+ B cells (**B**), and Gal-3 mRNA expression (**C**) in patients responding to treatment (complete or partial remission) and those non-responding (stable or progressive disease). The bar plot is used to show the median. “Whiskers” represent the lower (Q1) and upper (Q3) quartiles. The Mann–Whitney U test was used for the comparative analysis of variables. ns, not significant; * *p* < 0.05, ** *p* < 0.01.

**Figure 10 cells-13-00030-f010:**
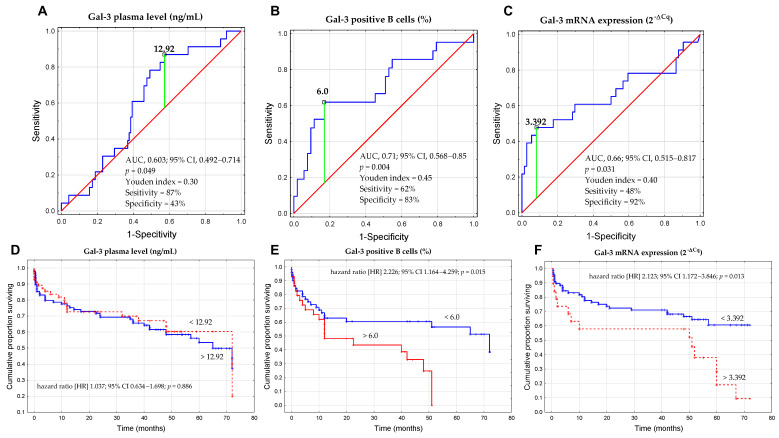
ROC analysis, AUC, and Youden index method were used to calculate the most significant cut-off value of the concentration of Gal-3 (**A**), percentage of Gal-3+CD19+ cells (**B**), and Gal-3 mRNA expression in B cells (**C**) that best distinguished the cases with del(17p) and/or del(11q). Kaplan–Meier curves for TTFT (time to first treatment) based on optimal cut-off for the concentration of Gal-3 (**D**), percentage of Gal-3+CD19+ cells (**E**), and Gal-3 mRNA expression in B cells (**F**). The TTFT was defined as the time from the date of diagnosis to the date of the initiation of the first treatment. ROC is the receiver operating characteristic; AUC is the area under the curve; HR is the hazard ratio; and CI is the confidence interval.

**Figure 11 cells-13-00030-f011:**
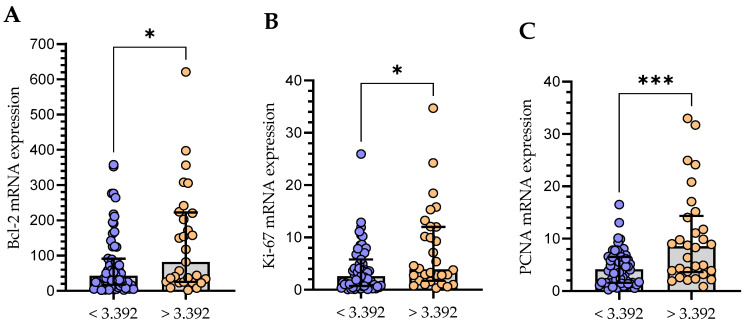
Comparison of Bcl-2 (**A**), Ki-67 (**B**), and PCNA (**C**) mRNA expression between groups with low (<3.392) and high (>3.392) Gal-3 expression. The bar plot is used to show the median. “Whiskers” represent the lower (Q1) and upper (Q3) quartiles. The Mann–Whitney U test was used for the comparative analysis of variables. * *p* < 0.05, *** *p* < 0.001; PCNA, proliferating cell nuclear antigen.

**Table 1 cells-13-00030-t001:** Characteristics of study and control groups.

Characteristics	CLL Patients (n = 146)	HV (n = 26)
Gender		
Female, n (%)	71 (48.6)	12 (46.2)
Male, n (%)	75 (51.4)	14 (56.8)
Age (years)		
Median (IQR)	66 (59–72)	58 (53–62)
Min–max	37–88	35–64
Rai Stage [24]		
0, n (%)	63 (43.2)	
I, n (%)	31 (21.2)	
II, n (%)	25 (17.1)	
III, n (%)	16 (11.0)	
IV, n (%)	11 (7.5)	
ZAP-70-positive, n (%) *	52 (35.6)	
CD38-positive, n (%) *	54 (37.0)	
IGHV mutational status		
U-IGHV, n (%)	22 (15.1)	
M-IGHV, n (%)	28 (19.2)	
not available, n (%)	96 (65.7)	
Chromosomal aberrations		
17p-, n (%)	12 (8.2)	
11q-, n (%)	13 (8.9)	
+12, n (%)	5 (3.4)	
sole 13q-, n (%)	40 (27.4)	
Patients treated during observation period, n (%) ^#^	58 (39.7)	
Complete remission (CR), n (%)	12/58 (20.7)	
Partial remission (PR), n (%)	28/58 (48.3)	
Stable disease (SD), n (%)	13/58 (22.4)	
Disease progression (PD), n (%)	5/58 (8.6)	
First line therapy		
fludarabine + cyclophosphamide + rituximab	10/58	
bendamustine + rituximab	15/58	
chlorambucil and obinutuzumab	14/58	
rituximab + cyclophosphamide + deksametazon	4/58	
venetoklax and obinutuzumab	8/58	
ibrutinib	4/58	
acalabrutinib	3/58	
WBC (G/L), median (IQR)	24.63 (17.2–55.16)	
Lymphocyte count (G/L), median (IQR)	18.32 (10.87–47.58)	
LDH level (IU/L), median (IQR)	349 (285–412)	
β2M level (mg/dL), median (IQR)	2.60 (2.08–3.40)	

* The ZAP-70 and CD38 molecules expression was assessed on CD19+/CD5+ cells. The cut-off points were 20% and 30%. ^#^ Evaluation of responses to anticancer therapy was made based on criteria of response to the treatment according to the International Workshop on Chronic Lymphocytic Leukemia (IWCLL) criteria [21]. HV, healthy volunteers; IGHV, immunoglobulin heavy chain variable gene; U-IGHV, unmutated IGHV; M-IGHV, mutated IGHV; WBC, white blood cell; LDH, lactate dehydrogenase; β2M, β2-microglobulin; IQR, interquartile range.

**Table 2 cells-13-00030-t002:** Univariate and multivariate analysis of TTFT.

	Univariate Analysis			Multivariate Analysis		
Risk Factors	HR	95% CI	*p*-Value	HR	95% CI	*p*-Value
High ZAP-70 expression	1.803	1.240–2.620	0.002	1.676	0.797–3.527	0.173
High CD38 expression	1.339	0.908–1.975	0.140	na		
High β2M level	2.437	1.631–3.643	<0.0001	2.319	1.103–4.875	0.026
Positive 17p− and/or 11q−	2.480	1.372–4.614	<0.001	3.038	1.343–6.871	0.007
High Gal-3 mRNA level	2.123	1.172–3.846	0.013	2.492	0.706–8.793	0.155
High percentage of Gal-3-positive B cells	2.226	1.164–4.259	0.015	0.696	0.220–2.205	0.538
High Gal-3 plasma level	1.037	0.634–1.698	0.886	na		

Variables with *p* < 0.05 in the univariate analysis were used for the multivariate analysis. HR, hazard ratio; 95% CI: 95% confidence interval; β2M, β2-microglobulin; na, not assessed.

**Table 3 cells-13-00030-t003:** Gal-3 expression in CLL patients over time. Each row in a table is a collection of data related to an individual patient.

	Membrane Gal-3 (%)		Intracellular Gal-3 (%)		Gal-3 mRNA (2^−ΔCq^)	
Patient No.	At the Time of Diagnosis	After Treatment	At the Time of Diagnosis	After Treatment	At the Time of Diagnosis	After Treatment
1.	2.89	2.86	2.59	3.39	1.75	3.78
2.	6.93	4.56	18.86	14.16	5.32	4.32
3.	8.96	5.41	7.66	6.02	3.80	3.63
4.	2.5	2.53	21.6	26.26	5.21	6.5
5.	4.1	8.06	23.75	26.34	2.58	3.22
6.	6.89	3.88	22.01	8.75	3.88	3.09
7.	3.02	4.02	9.89	11.82	3.21	4.58
8.	5.57	4.56	8.36	6.73	4.64	3.66
Median	4.835 *	4.29 *	14.38 ^#^	10.29 ^#^	3.84 ^^^	3.72 ^^^
IQR	2.92–6.92	3.12–5.19	7.83–21.91	6.19–23.24	2.74–5.07	3.32–4.51

Peripheral blood samples were isolated from CLL patients at 6 or 12 months after treatment. IQR, interquartile range; Wilcoxon’s matched-pairs rank test was used for the comparative analysis of variables. *^,#,^^ *p* < 0.01.

## Data Availability

The data presented in this study are available within the article. Other data that support the findings of this study are available upon request from the corresponding author.

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
