# Peer review of "Can Galectin-3 Be a Novel Biomarker in Chronic Lymphocytic Leukemia?"

_cells, 2023, doi:10.3390/cells13010030_

Round 1

Reviewer 1 Report

Comments and Suggestions for Authors

The manuscript entitled " Can Galectin-3 be a Novel Biomarker in Chronic Lymphocytic 2 Leukemia?" 

Although, most of the study results are negative, it add some novel information of biological and clinical value.

I have several comments and suggestions:

1)      Patient cohort: It is not clear if all blood samples were taken while patients were "treatment naïve"? And if the cohort is mixed both by treated and non- treated patients: how did you interpreted patients who achieved CR? (as they have no CLL cells )

2)      The study aim to analyze the prognostic role of GAL3: Do you have data regarding IGHV mutational status? This factor is required for calculation of CLLIPI and perform a correlation with the international prognostic scoring.

3)      It would be of major advantage if you can the expression of GAL3 in lymph nodes and bone marrow, as we know now that different compartments has different role in CLL development.

4)      Do you have longitudinal results of the same patient? Before and after treatment?

Author Response

Dear Reviewer, thank you very much for your invaluable suggestions. We did our best to address them all properly during the short time we were given for revisions. We hope you will agree that after implementing your suggestions and those of Reviewer 2, the overall quality of our paper increased significantly. Please find the more detailed answers to your suggestions and questions below.

  1. Patient cohort: It is not clear if all blood samples were taken while patients were "treatment naïve"? And if the cohort is mixed both by treated and non- treated patients: how did you interpreted patients who achieved CR? (as they have no CLL cells )

Response 1. Peripheral blood samples were collected at the time of diagnosis, before any anticancer therapy. All subjects were newly diagnosed and treatment-naive. Additionally, eight CLL patients requiring treatment were studied at two-time points: at the time of diagnosis (before the start of the treatment) and 6 or 12 months after therapy. The data for untreated and post-treatment CLL patients were not mixed. The information on patient treatments was provided in section 2.1 (lines: 82-84). Evaluation of responses to anticancer therapy was based on the criteria of treatment response according to the recommendations of the International Workshop on Chronic Lymphocytic Leukemia (IWCLL). In the revised version of our manuscript, appropriate information was added (footer of Table 1, lines 101-103).

  1. The study aim to analyze the prognostic role of GAL3: Do you have data regarding IGHV mutational status? This factor is required for calculation of CLLIPI and perform a correlation with the international prognostic scoring.

Response 2. IGHV mutation status was available only in 50 of 146 CLL patients (Table 1). Nonetheless, in a revised version of our manuscript, the comparison of Gal-3 expression between mutated-IGHV and unmutated-IGHV groups was added (lines: 274-280). Figure 6 was added. CLL-IPI combines 5 parameters: age, clinical stage, TP53 status [normal vs. del(17p) and/or TP53 mutation], IGHV mutational status, and serum β2-microglobulin. Unfortunately, IGHV mutation status was available only in 50 CLL patients. IGHV mutation testing is not routinely performed in our laboratory. So we couldn’t stratify all patients with CLL into four risk categories. Thank you very much for this suggestion. In future, we will try to supplement our results with the CLL-IPI index.

  1. It would be of major advantage if you can the expression of GAL3 in lymph nodes and bone marrow, as we know now that different compartments has different role in CLL development.

Response 3. In our study, blood samples were collected during routine diagnostic tests. The crucial role in CLL diagnosis is played by blood tests, so bone marrow aspirates and lymph node biopsy samples are generally not required for CLL diagnosis. As a result, we did not examine bone marrow and lymph node samples. The use of bone marrow or lymph nodes was not approved by the Ethics Committee of the Medical University of Lublin. The Reviewer's suggestion is a very good idea for our future research.

  1. Do you have longitudinal results of the same patient? Before and after treatment?

Response 4. Eight CLL patients requiring treatment were studied at two time-points: before treatment and 6 or 12 months after therapy. The Gal-3 expression was significantly higher before the initiation of therapy compared to the values after treatment. We added these results into a revised version of our manuscript (“Results” section; part: 3.3. ‘Assessment of the influence of Gal-3 expression on the clinical outcomes’, lines 364-368). Gal-3 expression assessed over time was presented in Table 3.

Reviewer 2 Report

Comments and Suggestions for Authors

Comments and Suggestions as attached

Author Response

Dear Reviewer, thank you very much for your invaluable suggestions. We did our best to address them all properly during the short time we were given for revisions. We hope that you will agree that after implementing your suggestions and those of Reviewer 1, the overall quality of our paper increased significantly. Please find the more detailed answers to your suggestions and questions below.

  1. Major

  1. Because the authors performed experiments in the bulk population of CLL patient’s cells to determine Gal-3 expression, it can easily fall to heterogenous Gal-3 expression. It is due to the diverse genotypes (like mutations or genetic aberrations), and clonal heterogeneity throughout B- CLL samples. Therefore, the authors should categorize B-CLL patient’s samples with relative homology in genotype in one group and determine Gal-3 expression in each group.

Response 1. In our study, all CLL patients were divided into three cytogenetic groups, taking into account the results of FISH studies. First group included patients without cytogenetic aberrations (n=76), second group - patients with 13q deletion as a sole abnormality (n=40) and third group - with  unfavorable genetic changes, i.e. 11q deletion, trisomy 12 and/or 17p deletion (n=30). Our results clearly show the highest levels of Gal-3 in the third group of CLL patients. Detailed information is presented in the Results section; part: 3.2 ‘The relationship between the molecular profile of patients and Gal-3 levels’, lines: 288-294).

  1. The research is lacking the validation of Gal-3 expression in protein level in cell lysate (cytoplasmic, nuclear, or whole cell lysate).

Response 2. In the revised version of our manuscript, multilabel confocal microscopy was performed on peripheral blood mononuclear cells using antibodies against CD19 and Gal-3 to visualize the intracellular localization of Gal-3 in CD19+ cells. It came to our attention that many antibodies that were useful in flow cytometry were also suitable in fluorescence microscopy. We performed some experiments where we took a portion of intracellular-stained cells for flow cytometry analysis and immobilized them onto microscopic slides using cytocentrifugation. After cells adhered to microscopic slides, DAPI was used to stain their nuclei. So basically, we combined immunofluorescence and flow cytometry, resulting in two different data formats from one experiment. (Materials and methods, 2.3. Confocal microscopy, lines: 126-138; Results, lines: 199-200, Figure 1 was changed and Figure 2 “Representative confocal microscopy images of Gal-3 of CLL patient” was added).

  1. The research is missing data in vitro in B-CLL cell lines or primary cells from transgenic mice bearing CLL. In these models, they should perform the techniques to have the genetic inactivation and overexpression of LGALS3 (which encodes Gal-3), to do the downstream experiments and analyses.

Response 3: Dear Reviewer, thank you very much for this valuable suggestion. We will seriously consider doing it (hopefully) in the near future. Unfortunately, we are unable to do it for the current revision. The use of mice was not approved by the Ethics Committee of the Medical University of Lublin.

  1. If the authors can have the in vitro model in comment #3, the in vivo study should be employed to characterize the influence of Gal-3 in therapeutic response in disease xenograft animals.

Response 4: The use of mice was not approved by the Ethics Committee of the Medical University of Lublin. The Reviewer's suggestion is a very good idea for our future research. Unfortunately, we are unable to do it during the current revision.

  1. In the Introduction, the authors should mention the frequency del(17p), del(11q) and/or 12 trisomy in B-CLL.

Response 5: In the Introduction (lines: 61-68) the sentences: ‘The most clinically significant markers of poor CLL outcome are the lack of mutations in the immunoglobulin heavy chain variable region (IGHV), chromosomal abnormalities, as well as high expression of ZAP-70 (zeta-associated protein 70) and CD38 (1–3). Chromosomal alterations in CLL are detected in up to 80% of patients. Among them, deletions of 11q (found in 5–20% of cases), 13q (found in more than 50% of CLL patients), 17p (3–8%), and trisomy 12 (10–20%) have a known prognostic value and play an important role in CLL pathogenesis and evolution, determining patients outcome and therapeutic strategies’ were added.

  1. Completely missing Table. 3 in the manuscript.

Response 6: We apologize for our mistake. All information on univariate and multivariate analysis is provided in Table 2. '3' has been changed to '2' (line 347).

  1. The Discussion is too long and could easily cause the tiredness of the readers. This is not a review paper; therefore, the authors must summarize and discuss their findings in shorter paragraphs.

Response 7: The Discussion section has been revised and shortened.

  1. Minor

  1. Line 32, what is “isolation”?

Response 1: 'isolation' has been replaced by 'following ex vivo culture’

  1. In introduction, line 35, 36 and 45, missing citations in each sentence.

Response 2: The citations have been added.

  1. In the materials, why did the authors not try to collect aspirated bone marrow from patients.

Response 2: In our study, blood samples were collected during routine diagnostic tests. The crucial role in CLL diagnosis is played by blood tests, so bone marrow aspirates and lymph node biopsy samples are generally not required for CLL diagnosis. As a result, we did not examine bone marrow and lymph node samples. The use of bone marrow or lymph nodes was not approved by the Ethics Committee of the Medical University of Lublin. The Reviewer's suggestion is a very good idea for our future research.

  1. In Table.1, “HV” should be written in full meaning for the 1st appearance.

Response 4: The abbreviation “HV” has been written in full meaning. (Line 80 and footer of Table 1)

  1. Missing annotation of “Fig. 1A” in the text.

Response 5: The reference to Figure 1A is in the line 116 and 194.

  1. In Fig. 3F, how can the authors explain the result showing statistical significance of Gal-3 mRNA expression in CD38+ B-CLL cells, comparable to HV.

Response 6: Increased CD38 expression on CLL cells is associated with aggressive disease. It is suggested that in CLL patients the CD38+ group is characterized by generally observed high Gal-3 expression. Line: 261-263

  1. Line 241, in which criteria did the authors know the patients require treatment therapy?

Response 7: The criteria of indications for treatment and response assessment proposed by the International Workshop on Chronic Lymphocytic Leukemia (IWCLL) were used. Line: 302-304

  1. Paragraph 251-258, the authors should show the details of treatment therapy, such as what kind of drugs or medications.

Response 8: Details of the treatment was provided in Table 1. The criteria of indications for treatment and response assessment proposed by the International Workshop on Chronic Lymphocytic Leukemia (IWCLL) were used (line: 302-304 )

Round 2

Reviewer 2 Report

Comments and Suggestions for Authors

Dear authors,

Thank you very much for your time to revise the manuscript and respond to my comments!

Although all the issues have been addressed and explained, I still have something not really satisfied when the in vivo experiments cannot be performed in the study due to the lack of Ethical Committee's Approval of authors' affiliation.

I can choose the Overall Recommendation as "Accept in present form". But I will leave the final decision for the Editor to consider, based on the criteria of the journal.

Thank you very much!